# Why Do Emergency Medical Service Employees (Not) Seek Organizational Help for Mental Health Support?: A Systematic Review

**DOI:** 10.3390/ijerph22040629

**Published:** 2025-04-17

**Authors:** Sasha Johnston, Polly Waite, Jasmine Laing, Layla Rashid, Abbie Wilkins, Chloe Hooper, Elizabeth Hindhaugh, Jennifer Wild

**Affiliations:** 1Department of Experimental Psychology, University of Oxford, Oxford OX1 2JD, UK; 2South Western Ambulance Service NHS Foundation Trust, Bristol BS16 IDE, UK; 3Phoenix Australia Centre for Posttraumatic Mental Health, University of Melbourne, Carlton, VIC 3053, Australia

**Keywords:** mental health, emergency medical services, paramedic, organizational support, barriers, enablers, ambulance, psychology, culture

## Abstract

Emergency medical service (EMS) ambulance employees play a critical role in emergency healthcare delivery. However, work-related experiences can compromise their mental health and job satisfaction. Despite available supportive services offered by EMS organizations, employee uptake remains low, while mental ill health and suicide rates remain higher than those of the general population. Understanding barriers to and enablers of such support is crucial for addressing factors that connect employees with the services designed to help. This systematic review identified 34 relevant articles and utilized an innovative process of integrating quantitative and qualitative aspects of the primary and gray literature to provide a qualitative synthesis of barriers and facilitators as perceived by EMS employees. Themes of employee (in)ability to ask for help, tailored person-centered support, and education and training about mental health were overarched by organizational culture. Barriers included perceived organizational obligation rather than genuine care, alongside machismo and stigma. Enablers included valuing and acknowledging employee risk by providing time and normalizing support utilization at work. Reframing machismo from dominance, competition, and toughness to respect, perseverance, and courage; promoting adaptive coping; and providing time and training were essential. Future research should aim to understand the factors influencing employee utilization of supportive interventions based on these themes.

## 1. Introduction

The World Economic Forum projected that more than half of the global economic burden attributable to non-communicable diseases will be caused by mental ill health by 2030 [1]. Emergency medical service (EMS) ambulance employees, at the front line of healthcare, are particularly vulnerable, facing unique challenges such as exposure to critical incidents, long hours, and shift work. Due to these factors, mental health and well-being can be compromised by work-related traumatic events, as well as everyday stressors caused by increasing demand for EMS services and wider healthcare system failures, such as lengthy hospital handover and discharge delays [2]. Consequently, EMS organizations struggle to provide psychologically safe workplaces that promote employee well-being and actively prevent harm.

Suicide is a particular concern, with fifty-five English EMS employee suicides registered with the Association of Ambulance Chief Executives suicide register since data collection commenced in August 2018 [3]. Furthermore, Mars et al. [4] and Office for National Statistics data [5] identified a 75% increased suicide risk among male paramedics compared with the general population. Several risk factors contribute to mental ill health such as genetics, circadian rhythm disruption, loneliness, stressful life events, and physical ill health. Among EMS employees, alcohol and drug misuse and high rates of childhood adverse events, such as trauma, abuse, and neglect, have been identified [6,7]. Research by the mental health charity Mind found that EMS employees were twice as likely to identify problems at work as the main cause of their mental ill health, compared with the general workforce [7]. Almost nine in ten EMS employees reported symptoms of mental ill health. Some EMS organizations reported 50% of employees leaving the workforce, compromising organizational capability, with poor staff mental health and organizational culture cited as primary contributing factors [8,9].

In response, EMS organizations provide employees with a range of supportive wellness services such as counseling, mindfulness, and referral pathways to charitable and community-based interventions. However, evidence suggests that the uptake of such services is low, and employee satisfaction surveys have not improved despite investment in supportive services [10]. Organizational support is a distinct concept that reflects an organization’s dedication to its employees. It includes the principles of effort–reward expectations, job satisfaction, and procedural fairness. EMS employee well-being is positively associated with access to organizational support, which is important as such support can reduce symptom severity, prevent suicide, and enable people to thrive at work [11]. Thriving employees underpin patient care and safety [12,13,14].

In the context of this review, organizational support refers to assistance provided by the employing EMS organization for employee mental health and well-being. This includes commissioned employee assistance programs (EAPs); peer support networks set up by the organization; and targeted interventions like post-incident debriefing, mindfulness classes, and counseling. Existing systematic reviews emphasize the importance of organizational support for employee well-being yet often lack detailed examination of the barriers and facilitators from an employee perspective. They also tend to exclude the gray literature, limiting insights from materials such as opinion pieces and industry briefings that capture nuances peer-reviewed research may overlook. Given the low uptake of support designed to aid mental health and well-being for EMS employees, this study aimed to gain a broader perspective on the factors influencing why employees will or will not utilize organizational support. To understand these factors, we expanded on prior reviews by integrating diverse sources and applying updated methodologies to identify underlying factors influencing support utilization. Our strategy adopted a comprehensive approach, while reducing bias, offering a more nuanced understanding of organizational support in this context. We conducted a full mixed methods systematic review of the literature, including all methodologies and gray literature. In addition, we identify supportive interventions offered through or by EMS organizations to support employee mental health and well-being reported in the included articles. Interventions are summarized using a Template for Intervention Description and Replication (TIDieR) checklist, which is presented alongside a comprehensive synthesis of the perceived barriers and enablers influencing whether employees will use the support provided.

## 2. Methods

A protocol for this review has been published [15] and registered with the International Prospective Register of Systematic Reviews (PROSPERO; CRD42022299650).

### 2.1. Search Strategy

Developed in partnership with expert librarians (see Appendix A), our search criteria included quantitative, qualitative, and mixed-method studies written in English since 1st December 2004. This date accounted for changes in employee well-being legislation and policy introduced to worldwide EMS organizations following the 11th September 2001 USA terrorist attacks [16,17,18]. The reference lists of selected studies were hand-searched for further material for inclusion.

Searches, using keywords and relevant thesauri, rerun in January 2024, included the following databases: AMED, CINAHL, the Cochrane Central Register of Controlled Trials and the Cochrane Database of Systematic Reviews via the Cochrane Library, EMBASE, EMCARE, HMIC, MEDLINE, PsycINFO, Scopus, and Web of Science. To ensure all the available and relevant research was captured, we sought gray literature from the OpenGrey, MedNar, and ProQuest databases and through the webpages of industry and charitable organizations active in supporting EMS ambulance employee mental health (Appendix A). We considered all article types, including systematic reviews and primary studies, to provide a comprehensive overview while avoiding data duplication by extracting and contextualizing the unique insights from each source. The search results are presented in a Preferred Reporting Items for Systematic Review and Meta-Analysis (PRISMA) flow diagram [19].

### 2.2. Eligibility Criteria

The searches were limited to organizational support for adults (18+) working for state/government commissioned out-of-hospital EMS ambulance organizations. We excluded EMS students, volunteers, friends or family, disaster response, intensive mobile units connected with specialist hospital services such as intensive care units, and private EMS workers as these groups were likely to be offered different or no support from EMS organizations. All EMS service job roles were included as evidence suggested that all could be affected by work-related experiences. We considered self-reported barriers to and facilitators of accessing and seeking organizational mental health support. Organizational support referred to assistance provided from EMS employers for employee mental health and well-being. This included commissioned EAPs; peer support networks set up by the organization; and targeted interventions such as post-incident debriefing, mindfulness classes, and counseling. We excluded informal social/family support, external agency support not commissioned by the employer (e.g., charitable organizations), and assistance from other health organizations such as general practitioners. Database papers were excluded if they (1) lacked relevance to the research question; (2) focused on patient- or population-based care; or (3) involved non-target personnel, such as in-hospital emergency department staff or police. Gray papers were primarily excluded for not addressing barriers to and facilitators of organizational support. For articles we could not access, we contacted the first authors using email addresses found through Google searches and www.researchgate.net.

Articles were imported into a Mendeley reference management system to remove duplicates and aid manual screening of titles and abstracts. One reviewer (SJ) independently reviewed all articles, and a second reviewer (AW) reviewed a random 10% sample (n = 479) from databases and registers and 10% (n = 18) from the gray literature. Both reviewers independently screened articles using a PICoT concept (see Appendix A).

Studies meeting all four criteria advanced to full text review and data extraction. Disagreements were arbitrated by a third reviewer (KS). Kappa statistics for inter-rater agreement demonstrated a moderate to good level of agreement (κ > 0.60) [20].

### 2.3. Quality Assessment

Quality assessment of all included papers was independently completed by two reviewers (SJ and LR) based on design and reporting. Different checklists were applied depending on the type of article and methodology being assessed (see Appendix B, Table A1, for checklist items) [21,22,23,24]. Each paper received a quality assessment score ranging from 0 (not reported) to 2 (fully reported).

Differences in checklist application or item assessment between reviewers were resolved through discussion and third-person arbitration if required. For instance, any studies reported as “mixed methods” but only utilizing cross-sectional surveys with brief free-text responses were assessed using a quantitative tool as the descriptive data were deemed to be insufficient for qualitative analysis. The results were summarized in a color-coded table to highlight whether articles were assessed as being high, medium, or low risk for bias.

### 2.4. Data Extraction

Underpinned by a realist approach and reflexive journaling (see Appendix A) we created and piloted a data extraction template in Microsoft Excel. Two reviewers (SJ and JL) independently extracted key data from the results sections of the included papers and quotations relating to barriers to and enablers of organizational mental health support. The two reviewers discussed, debated, and resolved data extraction decision-making during regular Microsoft Teams meetings, with a third reviewer (KS) available to arbitrate any unresolved issues. Due to the lack of research and the complexities underpinning EMS employee perceptions, an open-minded inductive approach was chosen over a theory-driven deductive method for generating and exploring EMS employee perspectives. The two reviewers compared, debated, and resolved differences in decision-making about data extraction during weekly meetings. In addition, a 12-item Template for Intervention Description and Replication (TIDieR) checklist was populated to summarize interventions discussed in the included articles and to contextualize the review findings [25,26,27]. If data were missing or additional information was required, authors were contacted.

### 2.5. Data Synthesis

Following the Joanna Briggs Institute (JBI) methodology, a mixed-methods enquiry was undertaken to capture employee barriers to and enablers of organizational support for employee mental health [28]. This involved “qualitization” by transforming quantitative data into textual descriptions to represent numerical data, which were subsequently integrated with qualitative data and uploaded to NVivo 12 software. Two reviewers using an inductive approach agreed on a coding structure. Reviewer 1 (SJ) coded all data, while reviewer 2 (JL) independently coded 10% for comparison.

Pragmatic selective coding relevant to barriers to and enablers of ambulance employees utilizing organizational mental health support adopted a predominately semantic focus, supported by latent coding when needed, to produce a realist, descriptive account of the findings from each of the included papers. The rationale for the construction of any latent coding was recorded to aid transparency and reflexivity and to minimize researcher bias. During weekly meetings and regular email, the reviewers ensured the developing coding structure was appropriate and made sense in the context of the data. Once all data had been coded, the final codes were reviewed and refined and nominal labels agreed upon. A reflexive thematic analysis of the data was then undertaken.

### 2.6. Data Analysis

Using reflexive thematic analysis (RTA), two reviewers (SJ and JL) collaboratively constructed data through shared screen Microsoft Teams calls and reflexively through independent journaling. Decision-making was compared and discussed, while building and revising themes, enabling bracketing and challenge of bias. Codes were crafted into categories that represented perceived barriers, enablers, or both, within the organizational support context, aiding systematic theme construction and development based on patterns for overlap and similarities. Visual thematic and conceptual mapping functionality in NVivo was used to explore relationships.

All themes were reviewed and agreed upon by three authors (SJ, JL, and PW) after confirming that all had a central organizing concept, distinct from each other and related to the aim of illustrating the lived experience and influencing factors for why ambulance employees would or would not seek organizational support for mental health. The revision of data ceased when existing themes were being continually re-identified, when all relevant data had been considered, and when the reviewers were satisfied with the thematic map.

### 2.7. Public and Patient Involvement and Engagement (PPIE)

The research protocol was rigorously reviewed by PPIE representatives. Seven representatives were patients or relatives of patients with lived experience of using emergency medical ambulance services. Additionally, a reference group of fifteen EMS employees from clinical and non-clinical roles, considered as PPIE for the purposes of this study, contributed valuable perspectives.

### 2.8. Sensitivity Analysis

The quality assessment of the included studies guided a sensitivity analysis, evaluating the robustness of the synthesis by determining whether excluding low-quality articles influenced the findings and our confidence in them [29,30]. Low-quality articles were identified by calculating a summary score for each paper, by summing the total score obtained across relevant quality checklist items and dividing by the total possible score (i.e., 28 − (number of “n/a” × 2)). For primary articles, we applied Kmet et al. [21]’s cutoff points, removing articles with a summary score below 0.55 to differentiate between higher- and lower-quality reporting. For the gray literature, known for inherently higher bias levels, we set a cutoff score of 0.75. The influence of removing these articles on the findings was assessed by evaluating the following:
Whether any codes were left without associated references;Any changes in the number and meaningfulness of references supporting the codes and assessing whether codes were still supported by the data;Any change in the distribution of codes associated with a theme and whether such themes were still representative of the data.

Following the completion of the sensitivity analysis, the reporting of this review was guided by the PRISMA Statement and the RTA Reporting Guideline (RTARG) [19,31].

## 3. Results

### 3.1. Search Results

From a total of 34 included papers, 23 articles [32,33,34,35,36,37,38,39,40,41,42,43,44,45,46,47,48,49,50,51,52,53,54] were identified from database and register searches and 11 articles [55,56,57,58,59,60,61,62,63,64,65] by other methods (see Appendix A). The articles represented a broad spectrum of study design and grey literature. Published research included 12 qualitative [32,33,34,36,37,38,45,49,51,53,60,63], 8 quantitative [41,43,44,47,50,56,58,61], and 6 mixed-methods studies [48,52,55,62,64,65], alongside 3 systematic reviews [39,54,66], and 1 randomized controlled trial [35]. Grey literature included 2 industry briefings [42,59], 1 quality improvement study [57], and 1 opinion piece [40]. Only one article, Clompus and Albarran’s [51] qualitative study, was forwarded to a third reviewer (KS) due to uncertainty about its relevance, who agreed it warranted inclusion. Data from eight of the included primary studies were also discussed within the three systematic reviews [33,36,37,38,45,51,57,60]. To manage information duplication, data from primary and systematic review studies were independently extracted and integrated during qualitative analysis. This approach strengthened results, as overlapping data encouraged reflexion and consideration of each source’s unique insights [28,67]. 

The included articles were published between 2007 and 2023, originated from 15 countries worldwide, included 20,354 EMS employees (alongside 169 papers included in Williams et al. [52] systematic review, where sample sizes were not reported). Sample sizes ranged from 6 to 4022. A total of n = 30 articles described supportive interventions offered by EMS organizations. Characterized using the TIDieR checklist, the most commonly reported interventions were post-incident debriefing, out-sourced EAP’s, and peer support networks (see Appendix A). The main characteristics of the included studies are presented in Appendix B, Table A2.

### 3.2. Quality Assessment Results

The quality assessment consensus is presented in Appendix A. Kappa statistics of reviewers’ agreement to checklist answers were calculated following binary categorization of answers into 1 = yes or partially reported and 0 = no, answer not reported. By comparing these binary responses, a kappa above 0.90 was determined, which is considered a very good level of agreement [20].

A total of n = 38 checklists were applied to the n = 34 studies (due to two checklists being applied to mixed-methods studies). The majority of studies were determined as low risk for bias (n = 18), with n = 8 studies assessed as unclear and n = 6 studies deemed high risk. Common methodological reporting issues were the lack of reflexivity in qualitative studies, poorly defined outcome measures in quantitative studies, and a lack of objectivity in gray articles.

### 3.3. Summary of Findings

The results were based on 102 codes systematically constructed from 34 studies between May and June 2024. Aided by NVivo software (see Appendix A), the codes were utilized to construct themes that represented data linked to employee perceptions of support delivery factors and organizational factors influencing the barriers to and enablers of utilization of organizational support for mental health. A number of data from quantitative studies were qualitized; e.g., frequency and percentage data from Ntatamala and Adams’ [47] quantitative survey were transformed into a textual statement of “Over a third of participants reporting barriers to help-seeking for work related stress, feared that services were not confidential, and this factor would be a barrier to utilizing support.”

Once the data had been qualitized, integrated, and thematically analyzed, the identified themes of employee (in)ability to ask for help, tailored person-centered support, and education and training about mental health were overarched by a central theme of organizational culture. This review reinforced existing knowledge that barriers include perceived stigma, fear that disclosure would negatively affect careers, and the importance of being provided with time at work, alongside peer and manager-led support for normalizing help-seeking behavior. We also identified new information that EMS employees perceive support as obligatory, rather than genuine, care from their employer, alongside machismo and the perceived importance of EMS context-specific and person-centered support options. These themes are presented in the context of barriers to and enablers of organizational support. An overview of the constructed themes and associated codes is presented in Appendix A, and illustrative quotes supporting the following themes are provided in Appendix A.

#### 3.3.1. Overarching Theme: Organizational Culture

Thirty-one articles discussed the ways in which employees perceived that EMS culture shaped how they interact, make decisions, and approach their mental health at work. Perceived culture underpinned the other three sub-themes by both hindering support utilization (due to resource limitations and cultural norms) and facilitating support utilization (when aligned with evidence-based interventions and cultural norms). Moreover, EMS employees work within a culture where organizational support was perceived as obligatory rather than genuine, leading to feelings of being expendable and undervalued.

Distressing experiences were frequently overlooked by the organization and were compounded by fear that disclosing mental health issues may result in adverse career implications [64]. Aggression and violence were viewed as simply part of the job, and those who could not accept this did not last long in the profession.

Inconsistent strategic commitment to employee well-being, poor communication, and stigmatizing leadership attitudes fostered an environment where mental health discussions were avoided. Machismo, characterized by an exaggerated sense of masculine pride, often linked to traits like dominance, aggression, and an aversion to showing vulnerability, hindered the acknowledgment of symptoms of mental ill health and help-seeking. In EMS, this reluctance, particularly among males, stemmed from perceived weakness, fear of judgment, and cultural expectations to “tough it out.” The need for belonging, togetherness, and camaraderie was prevalent but intertwined with concepts of “brotherhood” and “bravado”, where expressing emotion was seen as a weakness and even emasculating [54].

Female employees found this masculine culture challenging at times and reported having to adopt “masculine qualities” such as being ”loud, robust, and engaging in male-focused banter” in order to fit in. Furthermore, for those who were able to express their emotions, this was conflated with weakness, seen as a lack of resilience, and an indicator that they “shouldn’t be in the job” [64]. A culture where weakness was conflated with emotional expression was a common feature felt across the whole workforce [64]. When combined with perceptions that the organization may not take mental health seriously, EMS culture itself becomes a powerful barrier to employees disclosing symptoms of mental ill health and utilizing support [38,62]. Furthermore, a lack of an open culture contributed to a lack of empathy, understanding, and resentment between colleagues [33,55]. The importance of prioritizing employee mental health and organizational encouragement of self-care and emotional disclosure, with the delivery of patient care, was highlighted and encapsulated by one paramedic: “How can I offer support [---] when we can’t even take care of ourselves?” [41].

In summary, the everyday working habits of EMS employees were strongly influenced by organizational culture. There appeared to be a misalignment between the needs of employees and EMS culture, especially around communication and normalization of accessing support.

#### 3.3.2. Subtheme 1: Employee (In)Ability to Talk About Mental Health and Ask for Help

Thirty-two articles discussed whether individual employees would disclose feelings and ask for help if needed. We found that organizational structure may complicate the landscape with a pervasive stigma around mental health, discouraging open discourse and perpetuating a culture that values stoicism [66]. Fear of appearing weak when discussing the impact of work on mental health and concerns that self-care might conflict with assigned job responsibilities can deter employees from speaking up and reaching out for help [39].

A lack of time at work was a particular barrier to accessing support. It was perceived that integrating time at work into organizational strategy to enable employees to reflect upon their own experiences and needs would help employees to feel valued and provide opportunity to connect staff with services designed to help [64].

A perceived lack of confidentiality when disclosing symptoms of poor mental health to others within the organization was described [64]. EMS employees operate in a protocol-driven, patient-centered environment. The lack of encouragement, time, and self-care directives undermines confidence in discussing mental health and seeking help. EMS employees feel it is important to feel safe and genuinely heard and to be met with genuine concern and empathy when they ask for help [37].

Employees’ ability to talk about their mental health was influenced by pre-EMS life experiences, such as traumatic childhood events. Such experiences may be a driving altruistic factor as to why some join EMS in the first place but may leave some employees vulnerable to certain types of incidents or general psychological stressors and could contribute to difficulties in seeking support. It was clear that work-related incidents could affect employees differently [51]. Incidents involving children and suicide, especially the suicide of a colleague, were seen as high-risk events for worsening employee mental health, especially if the aftermath was poorly managed by the organization [62].

Posttraumatic stress disorder (PTSD) was a concern, and factors such as avoidant coping were discussed alongside delayed psychological impact following incidents. These factors may create barriers to utilizing support as employees may not be ready or able to discuss their experiences within the early weeks and months following an event, when commonly used interventions such as critical incident stress management (CISM) and trauma risk management (TRiM) interventions are often provided. Consequently, EMS systems should be designed to adapt and respond to individual needs and provide support on a person-centered basis [59].

Speaking with peers was viewed as beneficial due to shared humor and camaraderie, which help to build positive workplace relationships and support systems [49].

However, building trust takes time, and employees may hesitate to access organizational support, including formalized peer support networks, until trust is established. Therefore, those early in their careers may rely on informal support outside the organization [36,37]. In contrast, more experienced employees are less likely to share their feelings and experiences with friends and family as they do not want to burden them [36].

Consequently, longer service can facilitate support use as experienced employees were more likely to utilize organizational support compared with newer recruits, despite reporting that they found it less useful. However, mistrust of organizational processes hinders support utilization for all employees regardless of length of service. This combined with the fear of burdening family and friends may leave employees feeling like they have nowhere to turn. Moreover, experienced employees may feel less comfortable discussing their mental health with managers [34]. Hierarchy seemed to create barriers in both directions, leading to a tendency to withhold and avoid emotional interactions observed across different hierarchical levels [34].

In summary, a perceived lack of confidentiality and encouragement to talk about mental health and well-being, combined with a lack of genuine concern and knowledge about what support is available to help, hampered the ability of EMS employees to discuss their feelings and utilize support when required.

#### 3.3.3. Subtheme 2: Provision and Utilization of Person-Centered Support Tailored to the EMS Context

Thirty-three articles discussed the provision of support offered by EMS organizations. A lack of easily accessible, timely, and useful support; insufficient choice; and a shortage of options tailored to individual needs were identified. These shortcomings discouraged some staff from accessing support due to doubts about effectiveness and fear that support could be actively unhelpful. A lack of information about evidence-based interventions was identified. For example, post-incident debriefing was a commonly discussed concept for enabling EMS employees to talk about their experiences and associated emotion. However, opinions about the usefulness and safety of this approach were mixed; it was perceived as useful for encouraging discussion, while others felt uncomfortable or unprepared to share their experiences so soon after the incident [64]. Formal debriefing practices such as CISM and TRiM were discussed. Both approaches addressed traumatic experiences; CISM focused on immediate post-incident support using debriefing and defusing techniques with the aim of reducing the risk of psychological damage, and TRiM provided immediate and follow-up support, educating employees about trauma responses and providing peer-to-peer individual/group risk assessments, with the aim of spotting signs of distress through a structured process, which might otherwise go unnoticed. Despite evidence that such approaches provide a structured risk assessment, which may be useful for assessing psychological symptoms and helpful for reducing stigma associated with mental health, these concepts were discussed with concern about potential harm and embedding negative feelings that lead to resentment [40]. In contrast, engaging in formal peer support programs for mutual support and emotional expression about an event was viewed in a positive light [54].

An experienced employee reminisced about peer support during their early career, when shifts were equally split between time spent in the vehicle and at the station, which allowed time for training and peer-to-peer debriefing. Time at the station has been eroded over the past 20 years, and newer recruits now only know a constant state of busyness [49].

Organizational culture has not evolved to maintain these supportive units of time for peer-to-peer debriefing and destressing as part of a preventative and proactive strategy. Therefore, formal peer-to-peer interventions and networks set up by EMS organizations that make time for peer-to-peer voice at work were perceived as being important. This shift toward reduced time together at the station has created a system that now relies upon employees to pluck up the courage to speak out, rather than one where taking time to reflect and talk is normalized. Evidence suggests that relying on individuals to ask for help is not working, since employees want to be asked about their well-being [60]. When EMS organizations do not ask whether employees are okay or provide time and supportive interventions at work, employees may not speak up and utilize the available support. Preventative systems that look and listen for all employees, rather than standing by and offering reactive support for those already distressed, should be provided [59].

Accessibility of the support offered was a key factor, with processes to accessing support reported as being onerous and a significant barrier (and in the U.S., prohibitively expensive for some individuals when the organization did not pay for supportive services). The visibility of how support is accessed and stigma were also barriers as some support services required employees to make themselves publicly known to be needing support [53].

The speed at which support was provided was also discussed. EMS employees provide immediate response to emergencies and appeared to hold expectations of a quick response when support was requested for themselves [59]. When responses were not provided quickly, or in some cases not provided at all, this may lead to negative perceptions, feed dissatisfaction, and create barriers for future service utilization [65].

An interesting aspect was the importance of talking with those who understood the unique EMS context. Scenarios were discussed where employee assistance providers were left in tears after hearing the stories of EMS employees, and employees were left feeling unsupported following such interactions or felt their experience could not be fully understood by someone outside the profession [63].

Concern about competency and the additional burden associated with formal peer-to-peer support was discussed, although this was outweighed by the importance of support facilitated and designed by those who understood the context. A call for more mental health professionals familiar with the EMS field was made [64].

Employees wanted a range of useful and evidence-based interventions delivered by trained, confident, and non-judgmental individuals who understood the context. Strengthening employee perceptions and confidence in the effectiveness of the support offered would increase confidence in using services when required, and including cognitive reappraisal within supportive structures was identified as a useful approach for reframing negative narratives [33].

Optional and mandatory organizational mental health support provision for EMS employees following traumatic calls was discussed. Mandatory approaches were perceived as useful for reducing stigma and signaling organizational commitment to employee mental health and well-being. When EMS organizations mandated employees to attend supportive sessions, individuals were not forced to participate if they did not wish to, although the practice was seen as useful for connecting employees perceived to be resistant to supportive offers. Encouraging employees to attend in support of others was a suggested method for including individuals perceived as reluctant to attend for their own benefit [45]. However, evidence suggested that making participation with support mandatory may make it less effective [44]. There was no empirical evidence to support or to refute the effectiveness of mandating time for support in reducing barriers to participation. However, mandating supportive time following certain incidents that were deemed as high-risk, such as critical incidents involving children, may be beneficial.

Overall, the adequacy and timing of the organizational support provided to employees is crucial as evidence suggests that this influences service utilization; mediates employee outcomes; and, in turn, improves patient care: “…the feeling of an abundance or lack of support directly affects the quality of care delivered to the patient” [40].

#### 3.3.4. Subtheme 3: Education and Training

Twenty articles discussed barriers to and enablers of organizational support in the context of training and education. Working for EMS carried with it an assumption that employees, by the nature of their work, would be well trained in understanding mental health, identifying symptoms of poor mental health, and knowing how to seek help when needed. However, the evidence suggested that employees felt underprepared for the psychological challenges of their EMS role due to a lack of training and protocols [53,64].

Reactionary culture was cited as a barrier to knowing when to utilize support due to the lack of time at work for training and education about the recognition of symptoms in themselves and others. Furthermore, employees were not involved in decision-making about intervention development and what support is offered, which fed into previously discussed issues of not knowing what support is available and underconfidence in effectiveness. Active engagement of employees with the development of supportive interventions could reduce the barriers to utilization of services [65].

A paramedic discussed the importance of co-production by sharing their experience of how the fire service involved their employees in developing organizational goals, mission, and values and the benefits this had for creating a cohesive workforce [65]. This reinforced the importance of leadership, of including employees in the creation of education and training, and ensuring that managers are also well trained. The influence of manager support for employees was discussed across the included articles. Managers were often the first point of contact when employees were exhibiting symptoms of poor mental health or were seeking support, and how managers dealt with this interaction was pivotal to employee well-being and support utilization [34].

Providing managers with training that sensitizes them to the effects of stress and enables them to detect stress in others should be an essential part of a manager’s job role. Emotional awareness and preparedness training would ensure all employees are equipped to detect changes in mental health. When this is combined with organizational readiness to provide evidence-based training and education, the barriers for employees seeking help when needed are reduced [39].

Training should aim to build knowledge about and confidence in utilizing support and should include preventative activities such as psychoeducation, burnout prevention, stress management techniques, and healthy coping education. This includes a greater depth of understanding about maladaptive coping such as avoidant-coping and drug and alcohol misuse. Training about suicidal ideation and suicide is also important since despite the known elevated risk among EMS employees, there was a lack of prevention and postvention training and protocols [47]. This links back to a lack of evidence underpinning what EMS organizations should offer their employees, who should offer support, and when. Evaluating the effectiveness of any adopted approach is important, and the evidence suggested that this was not always thought about [64].

The mode of evaluation requires careful consideration. In particular, the limitations of survey methods for examining support service satisfaction among EMS employees were discussed, identifying that appreciation of a service does not always equate to service effectiveness. Furthermore, as reflected in previously discussed themes, employees are often mistrustful of organizational motives and worry that disclosure of feelings may be used against them. This may influence how truthfully employees respond to surveys [64].

Additionally, evaluation methods, such as evaluating services during annual support meetings, may be unhelpful, especially for ambulance stations with less adequately functioning support [45].

In summary, nurturing employees and organizational processes through education and training about recognition, disclosure, and support utilization for mental health were important factors in connecting employees with organizational support.

### 3.4. Sensitivity Analysis Results

Following quality assessment, 6 articles were identified as low-quality based upon the reporting in the included articles, 3 used a quantitative methodology and three were grey articles (3 from the USA, 2 from Canada, and 1 from the UK) [40,42,44,59,61,64]. The codes associated with these articles were removed from the data analysis using NVivo software, leading to a 25% reduction in the underpinning references of the 102 codes. The impact on the themes and underpinning codes were visually assessed by one author (SJ), reviewed and agreed by two others (JL, PW), and recorded in Appendix A.

Themes continued to be supported except for three factors. First, the concepts contained within the subtheme of “employee (in)ability to talk about mental health and ask for help” remained unchanged, despite being weakened by the removal of low-quality studies. The concept of engaging stakeholders to inform decision-making was left unsupported in the overarching “culture” theme. From the “training” theme, the concept of evaluating and measuring effectiveness of the support provided was left unsupported. Finally, the concept of inclusive support that was representative of the workforce, which was nested within the “support provision” theme, was also left unsupported.

These concepts are important, although it could be argued that from an employee lens, concepts such as effectiveness measures and engaging stakeholders may be unfamiliar. The concept of inclusivity is vital to organizational support; nevertheless, the loss of this concept following sensitivity analysis may be attributed to the lack of population representativeness among EMS populations and EMS-related research [68]. Overall, the sensitivity analysis demonstrated that the constructed themes were robust and reflected the voices found in the underlying data.

## 4. Discussion

This systematic review aimed to identify and synthesize what is known about the factors influencing why EMS employees will or will not utilize organizational support for mental health and well-being. We identified 34 articles from the published and gray literature and synthesized the findings to construct one overarching theme and three key subthemes, which align with and expand upon the existing literature. Sensitivity analysis identified that reporting the quality of the included articles did not significantly change the findings.

Perceived barriers included a perception that organizational efforts were obligatory rather than genuine, prevailing machismo, fear of stigma, negative career consequences, and distrust in organizational motives and processes. Additional obstacles included concerns about manager and colleague discretion and ability to support mental health, lack of time at work for support, and insufficient awareness of evidence-based resources and interventions.

Enablers involved belief in organizational values; a culture of self-care, which included time at work for support; proactive measures; and strong leadership commitment. Factors that can be either barriers to or enablers of help-seeking included pre-ambulance life experiences and the impact of work-related incidents, associated coping techniques, and how such experiences shaped attitude and confidence in utilizing support. These factors were identified alongside the quality of communication and support offered [6,60,65].

This review identified that culture lies at the heart of whether employees will or will not utilize support provided at work. Employees want to be proactively asked about their well-being, and if they share their feelings, they want this information to be met with genuine concern, discretion, and evidence-based interventions designed to help. EMS organizations consider employee mental health and well-being through formal norms, such as regulatory obligations and well-being policies. However, informal norms of team dynamics, leadership style, traditions, and rituals appear to be more influential in employee willingness to be open about and seek help for mental health and well-being [69]. This emphasizes the detrimental effects of perceived organizational indifference about employee well-being.

An important factor to consider when discussing organizational support is the variation in operational models across EMS organizations. Differences in staffing structures, service delivery expectations, and workload management can influence employee stress levels, coping mechanisms, and perceptions of organizational support. For instance, high-intensity, resource-limited environments may exacerbate employee distress, while findings from this review suggest that there is appetite for well-supported models with structured mental health initiatives, which may mitigate psychological strain [70].

EMS employees facing mental health challenges encounter several barriers to accessing organizational support regardless of their job role. The perception of organizational obligation, where support is seen as a duty rather than genuine care, can deter help-seeking as employees need genuine psychological safety to share their feelings. Consistent with Caesens et al. [71] and Brunetto et al. [72], feelings of expendability make employees hesitant to seek assistance, perceiving repercussions for doing so or developing perceptions of being deemed replaceable or as lacking aptitude for their role.

Evidence reinforces the importance of individuals taking responsibility for understanding, recognizing, and taking action to look after one’s own mental health and well-being [73]. However, tension between individual and organizational responsibility for well-being is well documented [48]. Therefore, employee-led strategy is insufficient as EMS organizational support policies rely on employees asking for help, despite evidence that self-recognition of mental ill health can be hampered by symptoms [74]. Disclosure of symptoms and the ability to ask for help can also be further compromised by cultural norms.

This is important as the machismo identified in this review may be a contributing factor to creating an environment where misogyny, known to permeate EMS culture, can grow as discussed in a recent review by the English National Freedom to Speak Up Guardian [75]. This report reinforces the view that not only are these concepts barriers to help-seeking, but they may be contributing factors as to why EMS employees need psychological help in the first place and why they are reluctant to utilize organizational support [75,76].

Employees perceived the culture to be overtly “macho” and stigmatizing toward seeking support for mental health, which, in turn, creates barriers to support utilization and aligns with the existing literature [77,78]. Machismo is often associated with male stereotypes that emphasize toughness, self-reliance, and dominance. Such traits may be desirable for coping with the unpredictable nature of emergency work. In EMS contexts, females may also exhibit traits associated with toxic masculinity; Clompus and Albarran [51] found that female paramedics reported adopting masculine qualities by becoming “geezer birds” to fit in.

Beyond individual perceptions, organizational culture plays a pivotal role. A stigmatizing culture can exacerbate barriers to organizational support. Factors such as insufficient time at work and funding for support programs, alongside inconsistent manager support, can further compound help-seeking barriers. Thus, understanding and addressing cultural aspects is essential, not only for ensuring that employees utilize support provided but for enhancing organizational capability, as perceptions of organizational support influence worker behavior and are vital for ensuring that employees feel valued and ready to perform to their best ability [79].

Enablers identified in this review shed light on the benefits of proactive approaches. Employees who believed in organizational goals and values were more likely to seek support. When organizational values aligned with employee well-being, a supportive environment was more likely to be fostered. This aligns with the known literature and resonates with principles of transformational leadership and positive organizational behavior as discussed by Avolio and Gardner [80] and Crawford et al. [81], as well as the EMS organizational briefings included in this review [42,59]. When well-being is prioritized within cultural norms, where preventative measures and stress education are emphasized, employees are more likely to engage with available support systems.

It was clear that genuine leadership commitment to mental health and well-being at work mattered to EMS employees. Leaders who actively supported employee well-being created an environment where seeking help was encouraged. The emphasis on leadership and organizational flexibility aligns with the findings of Neilsen et al. [82], whose systematic review identified that these factors are crucial for fostering a healthy work environment, allowing employees to balance work and personal needs, while reducing stress and enhancing overall health. Despite progress in developing operationally effective managers, EMS organizations still face challenges in cultivating supportive leaders capable of fostering an open culture and connecting employees with necessary support. Prioritizing employee well-being and fostering a supportive culture remains crucial for connecting employees with the support designed to help. This approach emphasizes the importance of organizational support planning that includes person-centered strategies [83]. Person-centered approaches could include connecting employees with experts in specialist services through general practitioners, charitable organizations, or other healthcare avenues.

It appears to be important to employees to be asked how they are (rather than a structured assessment) after a traumatic incident at work to improve support utilization and to ensure people feel valued by their colleagues and employer. Some employees reported difficulties with self-reporting symptoms and felt under-confident and apprehensive about reaching out to colleagues if they perceived them to be experiencing poor mental health.

This may lead to challenges in providing adequate support for employees who may under-report symptoms. Additionally, pre-ambulance adverse life experiences, such as exposure to early childhood traumatic events (often the driving factor for joining EMS), coupled with the cumulative impact of work-related incidents contributed to whether employees would or would not utilize the provided support [51,84,85]. Adverse childhood experiences can increase vulnerability following subsequent traumatic events and lead to more severe anxiety and depression in adulthood [86]. Employees easily triggered by events may experience stigma at work and questions about their suitability for their EMS role, which hampers their ability to speak up and seek help. Conversely, lived experience of adversity may enhance empathy and ability to connect with patients.

The literature emphasizes the value of experiential knowledge for improving care [87]. Those who were perceived as being more resilient due to their lived experiences may be perceived as being less caring or more prone to compassion fatigue [88,89,90]. Previous experiences of support can also inform decision-making about utilizing support, especially if previous experiences were negative or ineffective. Recognizing these complexities and adapting supportive interventions to accommodate differences in factors such as the psychological processing time, severity of post-incident symptoms, and type of support needed allows organizations to tailor support. This, in turn, will help to minimize negative influences of lived experience on clinical decision-making and support EMS organizational goals of reducing unwarranted variation in patient care [91].

The findings of this review underscore the value of trust in facilitating employees to speak about mental health and, in particular, with people who understand their working context. Reports of counselors becoming tearful upon hearing work-related stories and perceived disingenuous understanding of what employees may be experiencing were barriers to engaging with support. Employing experts or training peers who understand the EMS context appears to be a vital link in ensuring that the provided support is meaningful to employees.

### 4.1. Implications

The findings of this review have important implications for public health policy and practice. There is a need for targeted interventions to foster a more supportive and inclusive work environment. Reframing machismo in EMS settings could minimize the negative aspects and help to de-stigmatize help-seeking culture. For example, shifting the narrative from dominance, competition, and toughness to positive aspects of respect, perseverance, and courage could help to transform the culture [92]. This, in turn, could enable employees to seek support when needed.

This review highlights the critical role of leadership in shaping organizational culture and employee well-being. Strong leadership commitment to mental health initiatives, coupled with communication and adequate time and resources, can improve the utilization and perceived effectiveness of support programs. The provision of person-centered support that addresses the diverse needs of employees, including those arising from pre-ambulance life experiences and work-related incidents, may help improve overall mental health outcomes in EMS workforces.

### 4.2. Strengths and Limitations of the Study

The strengths of this review include examination of the gray literature. We also undertook a rigorous methodological approach to strengthen the reliability of the findings. We found the innovative method of integrating qualitative and quantitative data designed by the Joanna Briggs Institute worked well in combining mixed-methods data for analysis. The use of inductive reflexive thematic analysis then ensured that employee voices were captured authentically while minimizing bias. The sensitivity analysis furthered the robustness of results. By removing low-quality studies, we addressed issues of methodological quality and poor reporting while demonstrating the strength of overall conclusions. These methodological choices collectively minimized the common pitfalls associated with mixed-methods systematic reviews [93].

Despite its strengths, the review has limitations. Only one author examined all the literature, and despite a reflexive approach to challenge bias, a single reviewer may inadvertently overlook relevant studies or introduce bias due to individual perspectives or preferences. While this was in part mitigated by recording decision-making and a second reviewer examining 10% of the data, this partial involvement may not fully capture the nuances of all included studies. A more comprehensive dual-reviewer process would enhance reliability and minimize potential oversights. We also note the moderate kappa inter-rater agreement score identified at the article selection process stage, which was likely hampered by the broad eligibility criteria. The criteria were purposefully designed to capture all the available information yet consequently resulted in large volumes of information unrelated to the research question.

The inclusion criteria focus on government/state commissioned EMS ambulance systems may limit generalizability. In some countries, only voluntary or privately funded EMS services may be available, and excluding such services could miss valuable insights. The consideration of broader contexts would enhance external validity and would need to consider the route to support for employees in these services.

Finally, a methodology that includes qualitization of data and reflexive thematic analysis, while valuable for capturing employee voices, can be subjective. Although carefully crafted, the interpretation of these data may vary, potentially affecting the replicability and reliability of findings.

## 5. Conclusions

Barriers to organizational support include prevailing machismo, a perception that support is being offered out of obligation rather than genuine care, cultural stigma about employee mental health, a lack of trust, and inadequate evidence-based resources. Although employees bring their own barriers to the table through lived experience and pre-formed attitudes toward help-seeking, they perceive that barriers to utilizing organizational support are most likely to be reduced by fostering empathetic cultures that prioritize time, money, and resources for employee mental health, while actively encouraging support utilization.

Enablers involved belief in organizational values and a culture of self-care and looking out for others, alongside dedicated time at work for employee mental health and well-being. Furthermore, proactive measures and strong leadership commitment, providing employee awareness and improving employee confidence to speak up, and integrating holistic approaches to organizational support were perceived to enable support utilization.

Factors that can be either barriers or enablers include pre-ambulance life experiences, length of service, the impact of work-related incidents, and the quality of communication and support within the organization. It was also perceived to be important to involve employees and experts in decision-making about what, when, and by whom support should be provided.

Reframing aspects of machismo and providing time at work for proactive (rather than reactive) support for employee mental health and well-being based on a culture of trust and genuine care will likely reduce barriers to support utilization. The future of EMS depends on its workforce, and without cultural normalization of self-care and help-seeking, EMS services risk compromising employee and, in turn, patient outcomes [94].

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
