# Peer review of "Why Do Emergency Medical Service Employees (Not) Seek Organizational Help for Mental Health Support?: A Systematic Review"

_ijerph, 2025, doi:10.3390/ijerph22040629_

Round 1
Reviewer 1 Report (New Reviewer)
Comments and Suggestions for Authors
Please see my comments attached

Author Response
Please see the attachment

Reviewer 2 Report (New Reviewer)
Comments and Suggestions for Authors
Thank you to the authors for this nice piece of work. I enjoyed reading it. I have few suggestions to improve the manuscript:
- You mentioned the exclusion and inclusion in the introduction (Lines 37n and onward), and that shouldn’t be placed there and should be moved to the method with section 2.2. Plus, could you tell us the limitations of the available systematic reviews on this subject and the added value of your study?
- You mentioned “two reviewers (SJ, JL) independently extracted.” Please add how you resolved the issue in case both reviewers had different opinions.
- Please clarify if you used an inductive or deductive coding approach in the data synthesis.
- I would also discuss the impact of the cultural context on different EMS employees as factors influencing EMS employees' mental health situation.
- I believe the operational model difference between various EMS is an important factor to be discussed.
Author Response
Please see the attachment

This manuscript is a resubmission of an earlier submission. The following is a list of the peer review reports and author responses from that submission.
Round 1
Reviewer 1 Report
Comments and Suggestions for Authors
Line 217- "the manual compared" - is not clear/obvious define
line 219 Primary author- is the first author?
EMS - includes not only ambulance services but also other prehospital emergency medical services ( intensive mobile unit connected with ED/ICU from hospital)- I suggest to explain clear if the review includes only the articles relates to ambulance cervices or all EMS
line 297 -the workshop and its results should be describe in a separate article/paper. this systematic review could be structured only using the classic method
Results starting on line 455 should be better structured with less examples. The results should present well defined. Please avoid "some employees" and use figures ( data in % or numbers).
Author Response
|
Comments 1: Line 217- "the manual compared" - is not clear/obvious define |
|
Response 1: Thank you for pointing this out. We have changed the relevant sentence [tracked changes document] p.6, lines 251-252: The extracted data was then manually compared any differences in opinion discussed and resolved. The two reviewers compared, debated and resolved differences in decision-making about data extraction during weekly meetings. |
|
Comments 2: line 219 Primary author- is the first author? |
|
Response 2: Many thanks, you are correct, we are referring to the first author here and have changed the wording from ‘primary’ to ‘first’ on [tracked changes document] p.6, line 225. |
|
Comments 3: EMS - includes not only ambulance services but also other prehospital emergency medical services ( intensive mobile unit connected with ED/ICU from hospital)- I suggest to explain clear if the review includes only the articles relates to ambulance cervices or all EMS |
|
Response 3: Many thanks for picking up this important point. Line 42, when the term ‘EMS’ is first mentioned, we state that this relates to ‘Emergency Medical Service (EMS) ambulance employees’ but we appreciate that there may be different perceptions about what this may mean in practice. To clarify, we have added the following to Table 1:
*Employees could include paramedics, Emergency Medical Technicians, Emergency Care Assistants, EMS ambulance nurses and doctors, emergency medical number call center and dispatch employees, operational managers, support, and central function employees such as Human Resources and patient safety teams, as well as senior leadership who work for out-of-hospital emergency medical ambulance services.
AND [tracked changes document] p.3, lines 134 – 135 eligibility criteria
We excluded EMS students, volunteers, friends or family, disaster response, intensive mobile units connected with specialist hospital services such as Intensive Care Units, and private EMS workers, as these groups were likely to be offered different or no support from EMS organisations. |
|
Comments 4: line 297 -the workshop and its results should be describe in a separate article/paper. this systematic review could be structured only using the classic method |
|
Response 4: Many thanks, we appreciate this feedback. To align with reviewer 2’s request we have added further detail about the experience of PPIE representatives on [tracked changes document] p.7 lines 331 – 333 and p.8 lines 334 - 336 and in line with your helpful comment 4, we have removed the following paragraphs from p8 lines 336-361: The refinement of our research question was guided by the outcomes that held the greatest significance for these stakeholders. Patient representatives emphasized ad-dressing negative employee attitudes during patient interactions and the importance of the mental health of EMS employees in clinical decision-making, while EMS em-ployees highlighted reducing colleague suicide risk and promoting a stigma-free envi-ronment for seeking mental health support. These factors informed decision-making about identifying what the barriers and facilitators to support were, as thriving and well supported employees were felt to be key to patient and employee outcomes. To further validate the findings of this review, we conducted an online workshop involving four employees from a UK EMS organization and one patient and public representative from a PPIE reference group established to support our re-search. During this workshop, SJ presented the identified themes and underlying ele-ments. The discussion centered on ensuring that the theme and underpinning factors effectively conveyed the lived experiences and influencing factors related to EMS em-ployees’ decisions to seek or avoid organizational support for mental health. Notably, the group emphasized several critical factors from our review, including providing dedicated time during working hours for employee welfare, consideration of the moti-vation, training, and confidence in persons delivering support, offering choice to EMS employees, and allowing the content of dedicated time at work to be flexed to meet in-dividual needs. The importance of our findings was underscored by the lived experienced of PPIE representatives who recognized the influence of employee mental health on patient care. As a result, the group concluded that “employee care is patient care” and should be the cornerstone of EMS organizational strategy. To disseminate our results, PPIE representatives will share a video summarizing these themes to aid in the recruitment of EMS employees for further research in this area.
|
|
Comments 5: Results starting on line 455 should be better structured with less examples. The results should present well defined. Please avoid "some employees" and use figures ( data in % or numbers). |
|
Response 5: Thank you, we have undertaken an extensive revision of the results section in line with your guidance, the changes can be seen in the tracked changes document. |
Reviewer 2 Report
Comments and Suggestions for Authors
· In Section 2.9, you introduce the concept of using PPIE representatives. Can you tell us more about the roles/experiences of these members? For example, 6 representatives were used, 4 of whom had prior experience delivering prehospital care, and 2 of whom were EM physicians.
· In the latter half of the same section 2.9, you provide additional details that are helpful about the makeup of the online PPIE group.
· Following each step of exclusion is a bit hard in the write-up. What was the primary reason for manual exclusion of database studies (n=4,668) of most of the records (n=4,795)? Why were 8 database reports unable to be retrieved? What methods did you use to attempt to retrieve them (e.g., interlibrary loan, contacting authors)?
· See above question for identification of other studies using other methods, regarding the same concept of exclusion.
· In the Sensitivity Analysis Results, you state that results were consistent across “country of origin”. However, your sensitivity analysis specifically targeted quality of articles, not origin of articles. Without performing the same sensitivity analysis, I would be reticent to make this statement about similarity across countries.
· Inclusion of appendix A2 is a researcher’s dream come true; very thorough!
· Excellent use of reflexivity!
· Thorough use of quality assessment practices.
· Extensive and interesting discussion!
Author Response
|
Comments 1: In Section 2.9, you introduce the concept of using PPIE representatives. Can you tell us more about the roles/experiences of these members? For example, 6 representatives were used, 4 of whom had prior experience delivering prehospital care, and 2 of whom were EM physicians. |
|
Comments 2: In the latter half of the same section 2.9, you provide additional details that are helpful about the makeup of the online PPIE group |
|
Response to comments 1 & 2: Thank you for highlighting this important area. We have reduced section 2.9 by removing lines p8 lines 336-361 [tracked changes document] as suggested by reviewer 1 and added the detail requested about the characteristics of the PPIE representatives as follows: 2.9 Public and Patient Involvement and Engagement (PPIE) The research protocol was underwent rigorously reviewed , informed by insights from was underwent by PPIE representatives. Seven representatives were patients or relatives of patients with lived experience of using who have direct experience with emergency medical ambulance services. Additionally, a reference group of fifteen EMS employees from clinical and non-clinical roles, considered as PPIE for the purposes of this study, contributed valuable perspectives. |
|
Comments 3: Following each step of exclusion is a bit hard in the write-up. What was the primary reason for manual exclusion of database studies (n=4,668) of most of the records (n=4,795)? Why were 8 database reports unable to be retrieved? What methods did you use to attempt to retrieve them (e.g., interlibrary loan, contacting authors)? |
|
Response 3: Thank you for pointing this out. In an attempt to retrieve the 9 database searches where the article was not available, we sent emails directly to the first author using email addresses identified through google searches and messages sent through ResearchGate.net accounts if available. One author, Gallagher et al., shared their paper via ResearchGate following this request which was subsequently included in our review. We did not receive replies from the other authors. To provide further explanation and clarity we have added the following section to [tracked changes document] p.3 lines 145-146 and p.4 lines 147-151: Database papers were excluded if they: (1) lacked relevance to the research question, (2) focused on patient or population-based care, or (3) involved non-target personnel, such as healthcare or emergency service personnel outside our scope. For articles we could not access, we contacted the first authors using email addresses found through Google and researchgate.net. |
|
Comments 4: See above question for identification of other studies using other methods, regarding the same concept of exclusion. |
|
Response 4: Thank you for highlighting this, we have added the following sentence to [tracked changes document] p.4 lines 149-150: Database papers were excluded if they: (1) lacked relevance to the research question, (2) focused on patient or population-based care, or (3) involved non-target personnel, such as healthcare or emergency service personnel outside our scope. Grey papers were primarily excluded for not addressing barriers and facilitators to organizational support. For articles we could not access, we contacted the first authors using email addresses found through Google and researchgate.net. |
|
Comments 5: In the Sensitivity Analysis Results, you state that results were consistent across “country of origin”. However, your sensitivity analysis specifically targeted quality of articles, not origin of articles. Without performing the same sensitivity analysis, I would be reticent to make this statement about similarity across countries. |
|
Response 5: Thank you for pointing this out. Upon reflection , we appreciate that we may not have enough evidence to bolster this statement to a satisfactory degree and therefore we have removed the following from [tracked changes document] p.21 lines 1013 – 1016: The themes were constructed from data identified across all included studies and constituents were consistent regardless of country of origin or any other factor. This suggests that EMS employees face similar barriers and facilitators to organizational support regardless of which country they operate in. |
|
Comments 5: · Inclusion of appendix A2 is a researcher’s dream come true; very thorough! · Excellent use of reflexivity! · Thorough use of quality assessment practices. · Extensive and interesting discussion! |
|
Response 5: Thank you for taking the time to add these positive comments, it is appreciated. |